# Impact of industrial robot applications on global value chain participation of China manufacturing industry: Mediation effect based on product upgrading

Shuangzhi Zhang [ORCID]*

College of Teachers, Chengdu University, Chengdu, China

* zhangshuangzhi@cdu.edu.cn

**Data Availability Statement:** All relevant data are within the paper and its Supporting information files. The dataset of this paper has been uploaded as an attachment in the submission system and has been uploaded to the Dryad database. DOI:

## Abstract

Promoting the application of industrial robot (IR) is an important module for China to build core competitiveness, and it is also the main grasp of global value chain participation (GVCP). Using China manufacturing industry panel data from 2006–2014, working from the perspective of product upgrading, this paper empirically analyzes the impact of IR applications on GVCP. The empirical results show that IR applications weaken China' incentives to participate in global value chains (GVCs); this weakening effect is reflected in both forward and backward participation in GVCs. On the one hand, the mediation effect test results indicate that the product upgrading effect brought about by IR applications can help China achieves the import substitution of intermediate inputs and uses local intermediate inputs to produce exports. These steps would reduce the backward participation in GVCs. On the other hand, the localization of manufacturing can result in China losing the opportunity to export intermediate inputs to other economies, thus reducing the forward participation of GVCs. Of course, due to sample limitations, the research conclusions of this article are only applicable to interpreting the Chinese economy.

## Introduction

Global value chains (GVCs) are global cross-firm network organizations that connect the processes of production, distribution, recycling, and disposal to realize the value of goods or services. This involves the entire process, from the procurement and transportation of raw materials, production, and distribution of semi-finished and finished products, to final consumption, recycling and disposal. Global value chain participation (GVCP) is an important way to promote international trade expansion, productivity, and employment growth in economies [1, 2]. The GVCs theory believes that GVCP can be divided into forward and backward participation. Forward participation is when economies export intermediate inputs used by other economies to produce exports; backward participation is when economies import the intermediate inputs used to produce exports [3, 4].

10.5061/dryad.280gb5mvr URL: https://datadryad.org/stash/share/ojZHg4V2fxznQ402yT_jo6EkP4fFblF2co_d5iI0VJM.

**Funding:** This work was supported by The MOE (Ministry of Education in China) Project of Humanities and Social Sciences (Grant No. 22YJC630207); Key Project of the Vocational Education Research Center of Chengdu University (Grant No. 2022ZD03). The funders had no role in study design, data collection and analysis, decision to publish, or preparation of the manuscript.

**Competing interests:** The authors have declared that no competing interests exist.

Entering the digital economy, the application of new generation information technologies, such as the internet, big data, and artificial intelligence (AI), has significantly impacted GVCP [5, 6]. Some studies have shown that the development and application of information and communications technologies (ICTs) facilitate the GVCP of economies [7]. This is mainly because advances in ICTs create favorable conditions for developed economies to transfer certain stages of production and manufacturing to developing economies [2]. Ali and Gniniguè [8] used second generation panel data for 41 African countries from 1990 to 2018 and found that GVCP, digitalization, and renewable energy are key determinants of structural transformation in Africa. Thus, on the one hand, developed economies can export intermediate inputs by other economies for the production of exports. On the other hand, developing economies can import the intermediate inputs used for the production of exports. However, global trade has changed profoundly since the financial crisis, with trade becoming increasingly concentrated within regions, rather than across regions [2]. Some studies have also attributed the shortening and reflow effects of global value chains to the developing and applying of digital technologies [2, 9]. From the studies available in existing literature, there are no consistent findings on the impact of digital technologies on participation in GVCs.

Industrial robot (IR) as the main representative of AI, it not only continues to replace human jobs, but also changes the production organization and business management patterns worldwide, having a significant impact on the dynamic evolution of GVCs [10, 11], and thus there is literature that is dedicated to the impact of IR applications on participation in GVCs [7, 12, 13]. As with the impact of digital technology applications on GVCP, the existing literature on the impact of IR applications on GVCP is inconclusive. Stapleton [2] argued that IR applications and GVCP may have a rather complex relationship. Even so, this paper still attempts to answer why developed economies are taking back manufacturing segments that were initially outsourced in the context of deeper IR applications. In contrast, some of the typical labor-intensive manufacturing activities remain concentrated in emerging economies; these activities do not shift to economies with lower levels of development [2, 12]. For example, some labor-intensive industries in the eastern coastal areas of China have been transferred to the central and western regions, known as "make cage for bird". When IR applications can increase the added value of production processes, the production processes of China industries do not necessarily need to be transferred to other developing countries, but rather to different regions within the country.

The literature has shown that, IR applications lead to good corporate performance in terms of technological innovation and productivity improvements [10, 14–16]. These positive changes are reflected in product upgrades in companies, as evidenced by the improved product quality and technological content. Product upgrading will allow for an increase in the value added of manufacturing, thus giving economies an incentive to retrieve or retain manufacturing, and this will have two implications. On the one hand, product upgrading can help economies achieve the import substitution of intermediate inputs and use local intermediate inputs to produce exports, thus reducing backward participation in global value chains. On the other hand, the localization of manufacturing can cause an economy to lose the opportunity to export intermediate inputs to other economies, reducing the forward participation in GVCs. Product upgrading is an important channel for bridging IR applications and GVCP. Therefore, this paper will analyze the impact of IR applications on GVCP from the perspective of product upgrading.

China is an important player in GVCs. During the initial stage of China's economic development, due to China's advantageous labor costs, the manufacturing chain in developed countries kept shifting to China. At that time, China participated in GVCs mainly in the form of backward participation. Even with the increasing level of China's economic development,

backward participation is still an important way for China to participate in GVCs. At the same time, the level of China's forward participation in GVCs has been increasing. Some of the country's manufacturing links have gradually shifted to economies with lower development levels, and intermediate goods are being exported to other economies. As Stapleton [2] found, manufacturing activities that move outside of China tend to be closely linked to China's supply chains. China is a dynamic economy in both forward and backward participation in GVCs. On the other hand, IR is widely used in China. A study by Cheng et al. [17] shows that, in 2016, China became the country with the largest stock of robots, accounting for 19% of the total global inventory. From these two aspects, this paper plans to empirically test the impact of IR applications and product upgrading on GVCP, specifically by using in dustry level panel data from China.

Compared with previous studies, the marginal contribution of this paper is as follows: (i) In terms of basic regression, we use the panel data of China's manufacturing industry from 2006 to 2014 to empirically test the impact of IR applications on GVCP, which not only expands the research field of influencing factors of GVCP, but also provides new development thinking points for emerging economies to seize the opportunities of IR applications and improve their disadvantageous position in the GVCs. (ii) In terms of mediation effect, based on the perspective of product upgrading, this paper studies the impact mechanism of IR applications on China's manufacturing GVCP, and constructs a logical chain of "IR—product upgrading—GVCP", which provides a theoretical analysis for explaining the "manufacturing reshoring" in the current international division of labor system, and also deepens the understanding of China constructs a systemic circulation of domestic economy. (iii) The analysis of the effects of IR applications is becoming increasingly rich, but these literature mostly focuses on the impact on labor markets and economic growth [18, 19]. Although some literature also involves research on GVSs upgrading, upgrading and participation are two different academic concepts. On the basis of clarifying the differences between GVCP and global value chain upgrading, this paper innovatively uses panel data of Chinese manufacturing industry to conduct indepth research on IR applications and GVCP.

The follow up arrangement of this paper is as follows: the second part is literature review, introduce relevant existing literature; the third part is propose research hypotheses; the fourth part is study design, design empirical models based on research hypotheses; the fifth part is empirical results and analysis, report the test results of the research hypotheses in sequence; and the sixth part is research findings and discussion.

## Literature review

Regarding the participation of economies in GVCs, scholars have focused on the impact of factors such as economies' factor endowments, the level of economic development, trade costs, and the degree of openness to the outside world [7, 20, 21]. The birth and development of IT has led scholars to have a keen interest in studying emerging technologies and GVCs [2, 7, 9, 22]. In addition, with the deepening of AI applications, scholars have conducted extensive and indepth research on the economic performance assessment of AI applications. In the available literature, studies have focused on discussing the impact of AI applications on economic growth and labor markets [23]. Some literature has also focused on the impact of AI applications on key economic variables, such as technological innovation, firm performance, bank performance, and international trade [10, 16, 24–26]. But the impact of digital technology represented by AI is a double-edged sword, with both positive and negative aspects [27]. AI applications and the resulting pervasive digitalization of the innovation function have often been associated with increasing possibilities for search and recombination, this makes the

mechanism for generating results more complex [28]. As a result, these outcomes may not always be the initially planned desired outcomes.

IR integrate advanced technologies in the fields of machinery, electronics, sensors, wireless communication, voice recognition, image processing, and AI, considered as the main representative of AI. Therefore, the impact of IR applications on GVCs has naturally become a hot research topic in recent years. Existing literature has, on the one hand, studied the impact of IR applications on GVCP and on the other hand has discussed the impact of IR applications on the upgrading of GVCs.

From the current literature, one can easily argue that the impact of IR applications on GVCP is complex [2]. Some scholars have expressed concerns that IR applications can negatively affect GVCP. Rodrik [9] argued that robotics applications in developed countries could potentially exclude developing countries from GVCs. Artuc et al. [13] specifically studied the impact of robotics applications in developed countries on developing countries' imports and exports. The study shows that, in developed countries, a 10% increase in robotics intensity is associated with a 6.1% increase in those countries' imports from developing countries, and an 11.8% increase in exports to these countries, resulting in an overall 5.7% decrease in net sectoral imports from developing countries. Another study by Artuc et al. [29] showed that automation can increase the relative advantage of the United States in certain industries and reduce the demand for Mexican products. Marin [12] noted that, while some U.S. firms are moving their manufacturing operations in China back to the United States in the context of IR applications, there is no clear evidence from an overall perspective that IR applications lead to a shift in offshoring activities from developing to developed countries. However, Martin's conclusions have yet to be rigorously empirically tested. From a microfirm perspective, Stapleton [2] argued that global trade tends to be dominated by a few very large productive firms, whose automation is more likely to enhance participation in GVCs. Antràs [7] emphasized that, even though IR applications may not negatively impact GVCP, it is important to be wary that IR applications may exacerbate inequalities in the benefits of economies' participation in GVCs. This is especially the case for less developed economies.

In terms of research on IR applications and the upgrading of GVCs, current studies have largely affirmed the positive effects of IR applications on the upgrading of GVCs [30–32]. Based on an empirical study of the textile industry in 28 countries from 2010–2017, Ma et al. [31] found that automation technology in the textile industry can significantly drive the GVCs division of the labor status of a country's textile industry. Ali et al. [32] emphasized that digitalization and structural transformation enhance the environmental quality in sectoral value chain participation in Africa since they significantly reduce carbon dioxide ($CO_2$) emissions. Furthermore, Ali et al. [33] analyzed panel data from 112 developing countries from 1990 to 2018 and found that digitization is an effective way for developing countries to reduce $CO_2$ emissions in the GVCs. From a micro perspective of firms' cross country production, Szalavetz [14] found that IR applications drive the multinational firms' headquarters and the upgrading of their manufacturing subsidiaries' value chains. Still, significant differences can be found in the impact of their upgrading patterns. In addition, studies have also shown that there is country variability in the impact of IR applications on the upgrading of GVCs. The contribution of IR applications to the upgrading of GVCs is more prominent in developed economies than in developing economies [30].

As for the mechanisms of IR applications on GVCs participation and upgrading, existing literature has focused on the specific mechanisms of how IR applications affect GVCs' upgrading. More studies are required on the mechanisms of how IR applications affect GVCP. From the available studies, scholars have generally agreed that IR applications achieve GVCs' upgrading in economies by promoting technological innovation, productivity improvement,

human capital accumulation, and industrial structure optimization [30–32, 34]. Modgil et al. [35] explored the new crown pneumonia and how IR applications enhance the resilience of GVCs in the context of the global spread of the epidemic. The study indicates that IR applications enhance GVCs' resilience by facilitating flexible sourcing strategies and providing personalized solutions for upstream and downstream supply chain stakeholders.

Overall, more theoretical and empirical research needs to be conducted on the relationship between IR applications and GVCP [2]. Antràs [7] also pointed out that exploring the impact of IR applications on GVCP is more of an empirical issue. Therefore, this paper will develop an empirical test specifically to analyze the impact of IR applications on GVCP. Although the existing rich empirical studies on IR applications and GVCs upgrading also help to understand the relationship between IR applications and GVCs' participation, generally speaking, GVCP and upgrading are not equivalent. A high level of participation does not necessarily lead to the upgrading of GVCs; similarly, a low level of participation does not necessarily hinder GVCs' upgrading.

According to Humphrey and Schmitz [36], Hausmann et al. [37], and Wu et al. [38], product upgrading seems to be a key mechanism variable that links IR applications and GVCP. Product upgrading is a specialized economic concept that refers to the commercialization of product with innovative changes in technology, materials, and processes [39]. Technological innovation continues to create new product, which will not only changes the existing product supply situation, but also promotes the upgrading of market consumption demand [40]. So, product upgrading achieves product complexity and unit value enhancement in the process of improving existing products and launching new ones, thereby affecting the position of economies in the international division of labor system [37].

Therefore, on the basis of studying the impact of IR applications on GVCP of China manufacturing industry, this paper further discusses the impact mechanism between the two from the perspective of mediation effect of product upgrading. An attempt is also made to explain what this study sees as the reasons for a typical phenomenon, namely that developed economies are taking back manufacturing segments that were initially outsourced, and some of the typical labor intensive manufacturing activities are still concentrated in emerging economies and have not moved to economies with lower levels of development [2].

## Research hypothesis formulation

### IR Applications and GVCP

Generally speaking, developed economies have traditional leading advantages in research and development (R&D), design, marketing and other links, and are deeply embedded in GVCs mainly through forward participation. The application of IR in manufacturing will further accelerate the innovation of intermediate inputs in developed economies, continue to increase the import demand of other countries for intermediate inputs from developed economies for reprocessing and manufacturing, and further enhance the forward participation of developed economies in the GVCs. Of course, if developing economies also innovate intermediate inputs with the help of IR applications, the export market share of intermediate inputs in developed economies will be eroded by developing economies, and the forward participation of advanced economies may be reduced. At the same time, if developing economies use industrial robot applications to achieve significant intermediate input innovation, they can also export intermediate inputs to third countries for processing and manufacturing to produce final exports, which will increase forward participation in GVCs.

However, there may also be cases where IR applications can also bring intermediate input innovation in developing economies, due to the low quality of product innovation and the

objective existence of product leadership in developed economies Intermediate inputs from developing economies may be more difficult to import by other countries and sold primarily within their own economies, in which case forward participation in developing economies does not show an upward trend. Szalavetz's research [15] based on the Hungarian automotive industry also shows that the digital transformation of industries represented by IR has not brought product innovation to Hungarian foreign manufacturing subsidiaries, but has brought significant product innovation to the parent companies of these subsidiaries. Therefore, the application of IR does not necessarily promote the embedded in the GVCs in the previous participa-tory mode of developing economies, so as to achieve the climb to the upstream of the GVCs. Therefore, in view of the actual situation in China, this paper proposes the research hypothesis 1a:

*Hypothesis 1a. IR applications will reduce the forward GVCP of China's manufacturing industry.*

China has a huge market space, and its internal regional economic development presents obvious stepwise characteristics. When the application of IR can bring about the increase in added value of the manufacturing link, the manufacturing link of China's industry does not necessarily have to be transferred to countries with a lower level of economic development, but between different regions of the country. As Stapleton [2] observed, in the context of the rapid development of "machine substitution", labor-intensive manufacturing activities in emerging economies such as China are still concentrated locally, without large scale industrial transfer to less developed economies. At the same time, the benefits brought by the application of IR have also prompted China to be more willing to participate in the manufacturing of the GVCs.

Under this circumstance, as one of the important economies in the global IR applications, China's IR applications can promote product innovation and achieve import substitution of intermediate inputs, Lastochkina et al. [41] has also found that emerging economies such as Russia can help achieve import substitution strategies through the application of IR. Thus, when China engages more in manufacturing activities, the intermediate inputs needed to produce final exports are to some extent supplied locally rather than imported from other economies, which leads to a decline in China's backward participation in GVCs. In short, China continues to promote the digitization and intelligence of its manufacturing industry, continuously enhancing the added value of production links, promoting the gradient transfer of industrial chains between different regions in the country. Considering the objective fact that most of China's manufacturing industry has not shifted to other developing economies, but from the eastern coastal areas of China to the central and western regions, this paper puts forward the research hypothesis 1b:

*Hypothesis 1b. IR applications will reduce the backward GVCP of China's manufacturing industry.*

Based on the comprehensive research hypothesis 1a and 1b, this paper proposes research hypothesis 1c:

*Hypothesis 1c. IR applications will reduce the overall GVCP of China's manufacturing industry.*

## Analysis of the impact of IR applications and product upgrading on GVCP

**IR Applications and product upgrading.** Humphrey and Schmitz [36] defined product upgrading as production activities in which producers increase the technological content of their products or make their products significantly different from other products. Banga [22]

found, based on an empirical study of manufacturing firms in India, that firms' application of digital technologies helps to produce better and more complex products in GVCs. As a typical representative of the new generation of information technology, IR applications also impact product upgrading [42]. Hong et al. [43] pointed out, based on an empirical study of Chinese enterprises, that when the degree of robotics application in enterprises reaches a certain threshold value, those applications bring about a significant product quality upgrading effect for exports. As a general purpose technology, AI can penetrate all areas of enterprise development and bring about the intelligent development of enterprise value chain links, such as R&D, design, manufacturing, and marketing. Since IR integrates numerous AI subdivision technologies, so this paper discusses the specific impact path of IR applications on enterprise product upgrading from the perspective of the enterprise value chain.

Most R&D and design activities are essentially intellectual labor-intensive activities that require continuous search and experimentation by R&D personnel; the marginal costs of search and experimentation are also rising [10]. The high cost and long cycle time of enterprise product innovation in general makes it difficult to achieve rapid product upgrading. A study by Verganti et al. [44] stated that the enterprise application of IR enables organizations to overcome many limitations of human-intensive design processes. This is achieved by improving process scalability, expanding enterprise scope across traditional boundaries, enhancing dynamic enterprise learning and adaptation. Also, IR can rapidly analyze big data and understand potential relationships from that data, potentially reducing experimental uncertainty and making the learning process more efficient, thus triggering additional new products [10]. Based on a biopharmaceutical analysis, Paul et al. [45] found that the application of automation technology in drug development allows for quick search and trial quantification, leading to better product design solutions. The intelligent development of an R&D design can help accelerate the process of the rapid introduction of brand new products to the market. This, in turn, is conducive to the leapfrog upgrading of enterprise products.

The intelligent development of manufacturing has accelerated the development of new manufacturing organizations, such as smart factories and digital twin factories, which continuously amplify the economies of scale and scope effects of manufacturing. This not only can reduce products' production costs but can also enhance the added value of manufacturing links [15]. Tanaka et al. [46] proposed uncertainty and cost adjusted models of firms' input choices, where prediction errors lead to under or over investment. The study found that IR applications can help reduce forecast errors and optimize firms' input decisions to improve the productivity of products. Through digital technologies like deep learning, graph computing, and knowledge graphs, companies can unfold customer profiles more accurately, based on customer big data information. This allows the companies to more effectively understand customer preferences, better tailor their products and services to customer tastes and needs, and improve the effectiveness and relevance of marketing [47]. When launching new products or expanding product offerings, companies need to be more certain about what customers want and how customer preferences may change. Using automation technology to analyze customer data may enable companies to overcome this obstacle [48].

In short, IR provides conditions for continuous quality improvement and product innovation implementation [49]. Therefore, based on the above analysis, this paper pro-poses the research hypothesis 2:

*Hypothesis 2. IR applications will significantly contribute to firms' product upgrading from the perspective of their value chains.*

**Mediation effect of product upgrading.** An economy's GVCP can be divided into forward and backward participation. Forward participation refers to an economy exporting the intermediate inputs used by other economies to produce exports. Backward participation refers to importing the intermediate inputs used to produce products for export. In this part, based on the analysis of IR applications and product upgrading, this study will discuss the impact of IR applications on the forward and backward participation in GVCs from the perspective of product upgrading. Then, the relationship between IR applications and GVCP will be clarified.

In a traditional GVCs' division of labor system, as an economy develops, that economy will outsource or shift the manufacturing processes that do not have cost advantages and are low value added. This is done to focus on the production of in-termediate inputs with high value added, to achieve the climbing within the GVCs' division of labor position. However, IR applications will act as a labor substitution, especially at the lower end, and reduce labor costs in manufacturing while increasing the value-added in product production. This will weaken the incentive for economies to outsource or shift manufacturing segments. Some economies are even gradually retracting outsourced or shifted manufacturing segments, leading to a reconfiguration of the geographic organization of production [50, 51]. This partly explains the phenomenon observed by Stapleton [2], who found that developed economies are retracting manufacturing segments that were initially outsourced. Meanwhile, some of the typical labor-intensive manufacturing activities still need to be concentrated in emerging economies and are not shifting to economies with lower levels of development. The above changes brought about by IR applications, in the context of IR applications promoting product upgrading, will weaken the motivation of economies to participate in GVCs in two ways.

First, as IR applications promote product upgrading, this can gradually meet the demand for the intermediate inputs produced locally in the economy and enable import substitution. Lastochkina et al. [41], Bulatova and Amirova [52] all argued that, in Russia's digital economy, digital technologies such as AI, IR, IoT, and big data will play an important role in Russia's import substitution strategy, because these digital technologies help to promote the upgrading of the industrial sector. Thus, when the manufacturing chain stays local, more of the intermediate inputs needed to produce final exports will be supplied locally, rather than being imported from other economies. This will lead to a certain degree of decline in the backward participation in GVCs, slowing down the overall evolutionary trend of GVCs' division of labor, a contraction of GVCs toward the leading economies, and the rise of localization momentum.

Second, if an economy is in a position to outsource or shift its manufacturing link, it will be able to free up more resources for the upstream link of GVCs to engage in innovation and production of intermediate inputs. Then, the intermediate inputs will be exported to the economy that has taken over the initial economy's manufacturing link and participate in GVCs in a forward direction. However, the manufacturing link still needs to be shifted out. In that case, economies with lower development levels will need help generating more import demand for intermediate inputs. This will result in an economy losing the opportunity to export intermediate inputs to other economies, thus reducing the forward participation in GVCs. Based on the above analysis, this paper proposes:

*Hypothesis 3. IR applications can weaken the motivation of China manufacturing industry to participate in GVCs by promoting product upgrading.*

## Study design

### Empirical model building

**Empirical model building of IR applications and GVCP.** This article uses Eq (1) to analyze the impact of IR applications on the GVCP of China's manufacturing industry, as follows:

$$GVCP_{it} = \alpha_0 + \alpha_1 IR_{it} + \lambda CTR_{it} + \theta_i + \varphi_t + \varepsilon_{it} \tag{1}$$

In Eq (1), $i$ and $t$ represent manufacturing industry segment and year, respectively; GVCP is global value chain participation, which is the result variable; IR represents the degree of IR applications, which is the explanatory variable; CTR represents industry level control variables; $\theta$ and $\varphi$ represent industry fixed effects (Ind FE) and year fixed effects (Year FE), respectively; $\varepsilon$ is the random error term, and $\alpha_1$ is the estimated coefficient of interest. If $\alpha_1$ is significantly negative, this indicates that IR applications inhibit the industry's GVCP.

**Empirical model building of IR applications, product upgrading and GVCP.** Theoretical analysis suggests that IR applications bring about product upgrading and thus reduce the incentives for economies to engage in GVCP. To test whether product upgrading is a channel through which IR applications reduce GVCP, this paper based on MacKinnon et al. [53], Selig and Preacher [54], and Zhao et al. [55], constructs a mediation effect model, as shown in Eqs (1) to (3). The mediation effect model helps describe how one variable affects another variable through the first variable's effect on some intermediate variable [54].

Specifically, research hypothesis 2 is tested using Eq (2), while research hypothesis 3 needs to be tested using Eqs (1) to (3) simultaneously, as follows:

$$Upgrade_{it} = \beta_0 + \beta_1 IR_{it} + \lambda CTR_{it} + \theta_i + \varphi_t + \varepsilon_{it} \tag{2}$$

$$GVCP_{it} = \gamma_0 + \gamma_1 IR_{it} + \gamma_2 Upgrade_{it} + \lambda CTR_{it} + \theta_i + \varphi_t + \varepsilon_{it} \tag{3}$$

In Eqs (2) and (3), Upgrade indicates the product upgrading of the industry. If β1 and γ2 are significant, IR applications' impact on GVCP is at least partly through product upgrading. Of course, if at least one of β1 and γ2 is insignificant, according to MacKinnon et al. [53], the Sobel test needs to be performed [56, 57] with the test formula shown in Eq (4).

$$S_{\beta_1\gamma_2} = \sqrt{\hat{\beta}_1^2 S_{\gamma_2}^2 + \hat{\gamma}_2^2 S_{\beta_1}^2} \tag{4}$$

In Eq (4), $\hat{\beta}_1$ and $\hat{\gamma}_2$ denote the estimated values of $\beta_1$, and $\gamma_2$, and, $S_{\beta_1}$ and $S_{\gamma_2}$ are the standard errors of $\hat{\beta}_1$ and $\hat{\gamma}_2$, respectively.

According to Zhao et al. [55], the ratio of mediation effect to total effect is shown in Eq (5).

$$Mediating\ Effects = (\beta_1\gamma_2)/\alpha_1 \tag{5}$$

### Variable descriptions

**GVCP variable.** Against the backdrop of the deepening international division of labor, the traditional gross trade based on national borders and finished products can no longer accurately reflect the value added realized by each economy in each production chain from the GVCs perspective. For example, in the global production system of certain products, China is mainly involved in the labor-intensive production and assembly process. This leads to China having a large export value of finished goods in the traditional export trade statistics. Still, the value added realized in China needs to be bigger, accurately reflecting China's participation and division of labor in the GVCs. To address the drawbacks of traditional export trade statistics, the Organization for Economic Co-operation and Development (OECD) and the World

Trade Organization (WTO) have proposed the concept of value added trade; scholars have also proposed relevant value added trade accounting frameworks to decompose the total export value added [3, 58, 59]. This paper refers to Wang et al. [59] to decompose the value added exports of each industry in China. Wang et al. [59] decomposed the exports of an industry into the direct value-added exports of that industry and the value-added exports of the upstream industry embedded in the focal industry. Specifically, the value-added decomposition of exports consists of four components: (i) value added exports, (ii) value-added that is first exported and then returned to the country, (iii) value added abroad, and (iiii) pure double counting.

Based on the decomposition of industries' value-added exports, this paper measures the GVCP of Chinese industries using the GVCP measure proposed by Koopman et al. [4], as shown in Eq (6).

$$GVCP = \frac{IV}{E} + \frac{FV}{E} \tag{6}$$

In Eq (6), GVCP is the global value chain participation index; IV denotes the domestic value added in intermediate products processed in the importing country and then exported to third party countries; FV denotes the foreign value added in domestic exports; and E denotes total exports. The larger the index is, the higher is the industry's participation in GVCs. Further, this participation index can be decomposed into forward and backward participation, with IV/E being forward participation and FV/E being backward participation. Forward and backward participation can indicate the degree of participation of a country's industry in the upstream and downstream links of GVCs.

**IR applications variable.** According to Cheng et al. [17], Acemoglu and Restrepo [60], Dauth et al. [51], using the number of robots stocked or installed in an industry to measure the degree of IR applications in that industry is the main method. For example, in a study on the application of robots in Chinese industries, Cheng et al. [17] used the number of robots stocked in the industry to measure the degree of IR applications. In addition, Acemoglu and Restrepo [47] eliminated the effect of industry employment numbers when constructing metrics for robotics application. Therefore, this paper, following the methods of Cheng et al. [17] and Acemoglu and Restrepo [60], measures the degree of IR applications in the industry using the number of robots stocked per 1,000 people employed in the industry.

**Product upgrading variable.** Humphrey and Schmitz [36] found in their study of the embedding of GVCs that, as firms in developing countries further integrate into the global market, to remain profitable, producers must increase the technological content of their products or make their products significantly different from other products. This type of production activity by firms is called product upgrading. Accurately quantifying product upgrading is more difficult, because doing so involves both vertical and horizontal upgrading. Suitable alternative indicators must be found to measure product upgrading in a given industry. Export technological sophistication mainly emphasizes the difference in technological content between products. The term refers to the shift from simple to complex products, which implies an increase in the technological content of the products produced by firms [37]. In the context of GVCs research, one can measure the technical complexity of an industry's exports in product upgrading.

This paper first decomposes the export value-added of each industry in China's manufacturing sector from the perspective of backward linkage. The decomposition is based on the export value-added decomposition framework of Wang et al. [59], with the help of the 2016 version of the World Input Output Database (WIOD). Then, the export technological complexity of China's manufacturing subindustries is examined, based on the measures

proposed by Hausmann et al. [37], as shown in Eqs (7) and (8). Higher export technical complexity represents better product upgrading of the industry.

$$NEXPY_j = \sum_j \frac{vx_{ji}}{\sum_j vx_{ji}} NPRODY_j \tag{7}$$

$$NPRODY_j = \sum_i \frac{vx_{ji}/\sum_j vx_{ji}}{\sum_i vx_{ji}/\sum_j vx_{ji}} Y_i \tag{8}$$

In Eqs (7) and (8), $vx_{ji}$ represents the value-added of exports in industry $j$ in country $i$; $Y_i$ is the GDP per capita in country $i$ after purchasing power parity; $NEXPY_j$ represents the technical complexity of exports in the industry, and the larger the value is, the higher is the technical complexity of exports. In the empirical study, the calculated value of export technical complexity is taken as the natural logarithm.

**Control variables.** Antràs [7] argued that, within a given economy, firm size, productivity, and other characteristic variables can significantly impact GVCP. In addition, based on data from Latvia and Estonia, Benkovskis et al. [21] found that exports positively affect GVCP and productivity. Kersan-Škabić's [20] study pointed out that profit tax rates significantly negatively impact GVCP in European Union (EU) member states. Referring to the scholars mentioned in the above studies, this paper includes control variables in the empirical study: (i) average operating income per firm in the industry, (ii) labor productivity (natural logarithm of value added generated per 1,000 employed people), (iii) the value of export deliveries as a share of operating income, and (iiii) income tax payable as a share of total profits.

**Data sources.** This paper aims to analyze the impact of IR applications and product upgrading on GVCP using Chinese manufacturing industry level data. Therefore, data from two areas are used to measure the study variables. (i) The 2016 version of the World Input Output Database (WIOD) is used to calculate the GVCP and product upgrading in 19 manufacturing industries in China, from 2000–2014. It should be noted that this is the latest data available from the WIOD, Wu et al. [38] also used this data for research on industrial intelligence. (ii) Robotics data collected by the International Federation of Robotics (IFR) is used to measure the degree of IR applications. The robotics data contain details of robots' annual installation and inventory in 14 manufacturing industries in China, from 2006 to 2020. (iii) These control variables are measured using the Socio Economic Accounts (SEA) data from the WIOD and the China Statistical Yearbook for all years. The Chinese Statistical Yearbook does not report value-added data by industry, but the SEA reports value-added data by industry and year for major countries, including China. Since the WIOD is not consistent with the IFR industry classification, the IFR industry classification was used as the basis for data matching. In short, this paper obtains panel data samples of 13 manufacturing industries in China from 2006 to 2014 from the WIOD and the IFR database through industry type matching.

## Empirical results and analysis

### Descriptive analysis

Table 1 presents the number of robot stocks per 1,000 employed population, and the GVCP in the sample industries in China, in both 2006 and 2014. As seen from the table, from 2006 to 2014, the number of robots stocked per 1,000 people employed in all 13 sample industries shows a significant growth trend. This clearly indicates that the level of IR adoption in Chinese industries was continuously increasing between those two years. In 2014, the capital and

**Table 1. Descriptive analysis.**

| Industries | Robot stock per 1,000 employed persons | | GVCP | | Forward GVCP | | Backward GVCP | |
|---|---|---|---|---|---|---|---|---|
| | 2006 | 2014 | 2006 | 2014 | 2006 | 2014 | 2006 | 2014 |
| Food and beverages | 0.01 | 0.47 | 0.12 | 0.10 | 0.02 | 0.04 | 0.10 | 0.07 |
| Textiles | 0.00 | 0.02 | 0.22 | 0.16 | 0.06 | 0.07 | 0.16 | 0.09 |
| Wood and furniture | 0.00 | 0.03 | 0.28 | 0.25 | 0.15 | 0.15 | 0.13 | 0.10 |
| Paper | 0.00 | 0.08 | 0.30 | 0.28 | 0.16 | 0.16 | 0.14 | 0.11 |
| Pharmaceuticals, cosmetics | 0.00 | 0.09 | 0.20 | 0.17 | 0.09 | 0.09 | 0.12 | 0.08 |
| Other chemical products | 0.00 | 0.22 | 0.38 | 0.36 | 0.23 | 0.25 | 0.15 | 0.11 |
| Rubber and plastic products (non-automotive) | 0.91 | 5.54 | 0.35 | 0.30 | 0.16 | 0.18 | 0.18 | 0.12 |
| Glass, ceramics, stone, mineral products (non-automotive) | 0.00 | 0.19 | 0.24 | 0.21 | 0.09 | 0.09 | 0.15 | 0.12 |
| Basic metals | 0.00 | 0.38 | 0.39 | 0.35 | 0.27 | 0.22 | 0.12 | 0.13 |
| Metal products (non-automotive) | 0.09 | 3.72 | 0.39 | 0.31 | 0.10 | 0.11 | 0.30 | 0.20 |
| Industrial machinery | 0.00 | 0.58 | 0.30 | 0.26 | 0.10 | 0.11 | 0.20 | 0.15 |
| Electrical/electronics | 0.15 | 14.97 | 0.33 | 0.27 | 0.17 | 0.16 | 0.16 | 0.11 |
| Other vehicles | 0.01 | 0.68 | 0.26 | 0.21 | 0.06 | 0.06 | 0.20 | 0.15 |

Note: ①The unit of 'robot stock' is one robot, for example, the value of 0.01 in the food and beverages industry in 2006 represents a robot inventory of 0.01 units per 1,000 employed persons in2006. The larger the number, the higher the level of IR applications in the industry. ②The number range of GVCP is from 0 to 1, and there is no specific unit. The larger the number, the higher the GVCP representing the industry.

technology intensive industries, such as electrical/electronics, rubber and plastic products (non-automotive), and metal products (non-automotive) were using more intelligent technologies, a finding that is broadly consistent with that of Cheng et al. [17]. In terms of the trend of GVCP, all the sample industries showed different degrees of decline in GVCP between 2006 and 2014. Of course, after distinguishing between forward and backward GVCP, it was found that there is a slight upward trend in forward GVCP in some industries, which may be related to the technological level of the industry. Mamba and Balaki [61] believe that GVCP has a clear time trend, which also suggests that time fixed effects should be controlled in regression analysis to obtain more robust regression results. The descriptive analysis in Table 1 suggests that IR applications reduced the dynamics of China's GVCP, although any rigorous conclusions will depend on the following econometric tests.

Of course, before econometric analysis, it is necessary to carry out diagnostic tests on variables, such as correlation, multicollinearity and panel unit-root tests. (i) Through the calculation of Pearson correlation coefficient, it was found that there were no values exceeding 0.40 in the three samples of GVCP, forward GVCP, and back GVCP, indicating a weak correlation between the variables. (ii) Through the calculation of the variance inflation factor (VIF) coefficient, it is found that there is no more than 2.00 values in GVCP, forward GVCP, and backward GVCP, which indicates that there is no multicollinearity problem between variables. (iii) Levin, Lin and Chu [62] introduced high-order differential lag terms on the basis of a universal panel autoregressive model to test the existence of unit-root in panel data, also known as the Levin, Lin and Chu (LLC) test. This paper conducts LLC test on the panel unit-root of GVCP, forward GVCP, and backward GVCP, and finds that these adjusted $t_{\delta}^{*}$ statistics are significantly negative, which means strongly rejecting the original assumption that the panel contains Root of unity and accepting the conclusion that the panel is a Stationary process. Further, after subtracting the cross-sectional mean from Panel data, the LLC test was conducted to avoid the possible crosssectional correlation effects of the disturbance terms in different industries. The results showed that these adjusted $t_{\delta}^{*}$ statistics are still significantly negative at the 1% level.

**Table 2. Regression results of research hypothesis 1.**

|  | (1) | (2) | (3) |
|---|---|---|---|
|  | GVCP | Forward GVCP | Backward GVCP |
| IR | -0.0027*** | -0.0014*** | -0.0013* |
|  | (0.0007) | (0.0005) | (0.0007) |
| CTR | YES | YES | YES |
| Ind FE | YES | YES | YES |
| Year FE | YES | YES | YES |
| R-squared | 0.9923 | 0.9819 | 0.9552 |
| Observations | 117 | 117 | 117 |

Note

***$p < 0.01$

**$p < 0.05$, and *$p < 0.1$; Robust standard error in parentheses.

**Testing research IR applications and GVCP.** Based on the data of 13 Chinese manufacturing industries, from 2006–2014, and according to the empirical model shown in Eq (1), this paper estimates the impact of IR applications on GVCP using ordinary least squares. The estimation results are shown in Table 2. As shown in Column (1), the estimated coefficient of IR applications is negative and passes the 1% significance test, indicating that IR applications reduce the GVCP of Chinese industries, the preliminarily proves the research hypothesis1c presented in this paper. Since GVCP can be divided into forward and backward participation, Columns (2) and (3) test the effects of IR applications on both forward and backward GVCP. From the estimation results, what is clear is that IR applications significantly reduce both forward and backward GVCP. It can be seen that the research hypotheses 1b and 1c have also been preliminarily verified. The empirical analysis (based on China's experience) shows that IR applications are an essential factor in reducing the country's GVCP dynamics. However, whether the above research conclu-sions are credible requires further robustness testing before it can be judged.

## Robustness testing IR applications and GVCP

**Replacing the IR applications variable.** Acemoglu and Restrepo [60] and Benmelech and Zator [63] use the installed base of robots to measure the degree of IR adoption in the relevant industry. Given that stock and installation are different conceptual categories, the degree of AI adoption in the industry is remeasured here using robot installations per 1,000 employed people to test the robustness of the study's findings. The regression results after replacing the IR adoption metric are shown in Table 3. The results show that IR applications significantly inhibits the rise of Chinese industry's GVCP; the inhibitory effect is also reflected in both the forward and backward participation.

**Instrumental variable regression.** The sources of endogeneity problems mainly include: sample selection error, measurement error of variables, omission of explanatory variables, causal relationship between the dependent variable and the explanatory variable, deviation of dynamic panels, etc. The existence of a two-way causal relationship between IR applications and GVCP, as in dicated by Beach [64] and Artuc et al. [13]. Drawing on Acemoglu and Restrepo [60], this paper constructs instrumental variables by selecting robotics data from countries with trends in robotics adoption that are similar to China's over the same period. The logic inherent in this choice of instrumental variables is that robotics, one of the most widely used types of automation technologies today, has become a globalized phenomenon in

**Table 3. Regression results for replacing the IR applications variable.**

|  | (1) | (2) | (3) |
|---|---|---|---|
|  | GVCP | Forward GVCP | Backward GVCP |
| IR | -0.0104*** | -0.0063*** | -0.0041* |
|  | (0.0023) | (0.0020) | (0.0025) |
| CTR | YES | YES | YES |
| Ind FE | YES | YES | YES |
| Year FE | YES | YES | YES |
| R-squared | 0.9931 | 0.9826 | 0.9551 |
| Observations | 117 | 117 | 117 |

Note

***$p < 0.01$

**$p < 0.05$, and *$p < 0.1$; Robust standard error in parentheses.

terms of both development and use. Robotics and related industries in any given country typically face strong competition from other countries; the help of robots in various industries in one country is typically strongly correlated with the use of robots in other countries. At the same time, no clear evidence exists that China's GVCs participation affects robot use in other countries.

As shown in Fig 1, while the U.S. robotics adoption level was ahead of China during the sample period, America's development trend was relatively close to that of China during the same period. In addition, according to Cheng et al. [17], with regard to the industry distribution of robot stock, one can see that the industry distribution of China's robot stock is most similar to that of the United States, as compared, for example, to Japan, Germany, and South Korea. Therefore, our selected instrumental variable for the application of IR in Chinese industries is the number of robots stocked per 1,000 employed population in the related industries in the United States.

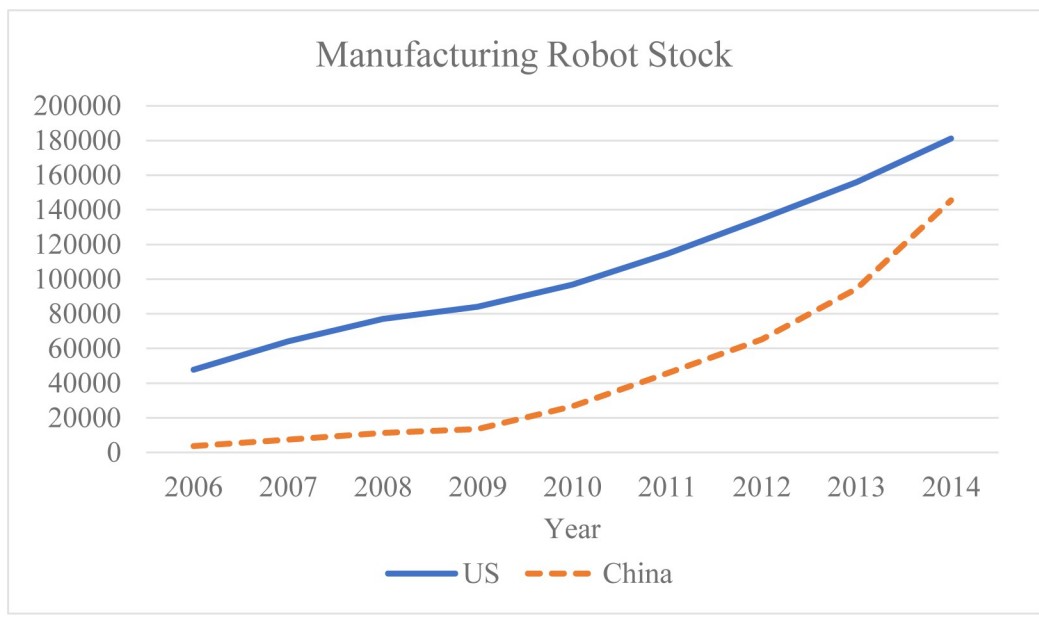

**Fig 1. Trends in manufacturing robot stock in the US and China.**

**Table 4. Regression results of instrumental variables.**

| | Manufacturing robot stock in US | | | One-period lagged variable of IR | | |
|---|---|---|---|---|---|---|
| | (1) | (2) | (3) | (4) | (5) | (6) |
| | GVCP | Forward GVCP | Backward GVCP | GVCP | Forward GVCP | Backward GVCP |
| IR | -0.0005*** | -0.0003*** | -0.0002* | | | |
| | (0.0001) | (0.0001) | (0.0001) | | | |
| IR_1 | | | | -0.0026*** | -0.0014** | -0.0012* |
| | | | | (0.0000) | (0.0006) | (0.0007) |
| CTR | YES | YES | YES | YES | YES | YES |
| Ind FE | YES | YES | YES | YES | YES | YES |
| Year FE | YES | YES | YES | YES | YES | YES |
| Kleibergen-Paap rk Wald F statistic | 8.4780*** | 8.4780*** | 8.4780*** | | | |
| Kleibergen-Paap rk LM statistic | 28.5590*** | 28.5590*** | 28.5590*** | | | |
| R-squared | 0.1765 | 0.0649 | 0.0402 | 0.9929 | 0.9837 | 0.9591 |
| Observations | 117 | 117 | 117 | 104 | 104 | 104 |

Note

***$p < 0.01$

**$p < 0.05$, and *$p < 0.1$; Robust standard error in parentheses. The weakly identified test refers to the Kleibergen-Paap rk Wald F statistic of the weakly identified test, and the original hypothesis of the test is that there is a weakly identified problem in the regression of instrumental variables; the unidentified test refers to the Kleibergen-Paap rk LM statistic of the unidentified test, and the original hypothesis of the test is that there is an unidentified problem in the regression of instrumental variables.

Mamba and Ali [65] used the instrumental variables approach to control the cross-sectional dependence and endogeneity issues. Referring to this method, this article selects the number of robots stocked per 1,000 employed population in the related industries in the United States as the instrumental variable and recalculates the impact of IR applications on GVCP using the two-stage least squares (2SLS) model. As shown in Table 4, Columns (1) to (3) display the regression results based on the instrumental variables. As can be seen from the table, both the unidentified test (Kleibergen-Paap rk LM statistic) and the weakly identified test (Kleibergen-Paap rk Wald F statistic) pass the sig-nificance test. These findings indicate that the instrumental variable in this paper do not has unidentified and weakly identified problems, and the instrumental variable has been effectively selected.

In addition, drawing on Groves et al. [66], this paper uses the one-period lagged variable of IR as the instrumental variable to re-analysis Eq (1). The regression results of the one-period lagged variable are reported in Columns (4) to (6). In Table 4, the estimated coefficients of IR applications are all negative and passed the significance test. This indicates that the conclusion that IR applications inhibit China's GVCP still holds after considering the endogeneity issue.

**Analysis of technological heterogeneity.** It is worth further exploring whether there is a correlation between the basic regression results and the intensity of industrial technology level. To verify this relationship, this article intends to study the relationship between IR applications and GVCP in different technology intensive industries in a Chinese sample.

Regarding the classification of high-tech manufacturing industries, this article refers to the classification standards of the National Bureau of Statistics of China and the OECD. In the "Classification of High-tech Industries (Manufacturing Industry)" of the National Bureau of Statistics of China (2017), high-tech manufacturing includes six industries: pharmaceutical manufacturing, aviation, spacecraft and equipment manufacturing, electronic and communi-cation equipment manufacturing, computer and office equipment manufacturing, medical instrument and meter manufacturing, and information chemical manufacturing. According

Table 5. Classification of high and low tech manufacturing industries.

| | (1) | (2) | (3) |
|---|---|---|---|
| | **Classification of High tech Industries (Manufacturing) by the National Bureau of Statistics of China (2017)** | **Adjust according to OECD classification** | **Classification of this article** |
| High-tech manufacturing industry | •Pharmaceutical manufacturing<br>•Aviation, spacecraft, and equipment manufacturing<br>•Manufacturing of electronic and communication equipment<br>•Computer and office equipment manufacturing<br>•Medical equipment and instrument manufacturing<br>•Information Chemicals Manufacturing | •Machinery and transportation equipment<br>•Chemicals<br>•Electronic and optical products | •Pharmaceuticals, cosmetics<br>•Other chemical products<br>•Rubber and plastic products (non-automotive)<br>•Basic metals<br>•Metal products (non-automotive)<br>•Industrial machinery<br>•Electrical/electronics<br>•Other vehicles |
| Low-tech manufacturing industry 5 | — | •Textile<br>•Leather Footwear<br>•Wood and wooden products<br>•Pulp and paper products coal<br>•Refining and nuclear fuel<br>•Rubber and plastic products<br>•Non-metallic mineral products<br>•Metal products | •Food and beverages<br>•Textiles<br>•Wood and furniture<br>•Paper<br>•Glass, ceramics, stone, mineral products (non-automotive) |

**Source**: Compiled by the author.

to the density of technology, the OECD divides the manufacturing industry into four categories: low-tech, medium low-tech, medium high-tech, and high-tech. Based on the industry classification requirements of this article, low-tech and medium low-tech are classified as low-tech, medium high-tech and high-tech are classified as high-tech. The specific adjustment of the OECD industry classification is shown in column (2) of Table 5. Finally, based on the classification standards of the National Bureau of Statistics of China and OECD, as well as the industry samples in this paper, the specific classification of high-tech and low-tech manufacturing industry is shown in column (3) of Table 5.

This article uses binary dummy variables to distinguish between high and low tech manufacturing industries, where "1" represents high-tech manufacturing and "0" represents low-tech manufacturing. To test whether the basic regression results will be affected by different technological levels in the manufacturing industry, this paper refers to the research methods of Mamba and Balaki [67] and introduces an interaction model for analysis. By setting the intersection term between IR applications and high-tech / low-tech manufacturing virtual variables as the explanatory variable, Eq (1) is regressed again to obtain the regression results shown in Table 6. As shown in Column (1), the estimated coefficient of IR*technology is negative and passes the 1% significance test, indicating that IR applications in both high-tech and low-tech industries have a restraining effect on the GVCP of China's manufacturing industry. Since GVCP can be divided into forward and backward participation, Columns (2) and (3) test the effects of IR applications on both forward and backward GVCP. The empirical results indicate that IR applications have a significant inhibitory effect on the forward and backward GVCP of China's manufacturing industry in both high-tech and low-tech industries.

**Testing research IR applications, product upgrading and GVCP.** Based on the research hypothesis 1, this paper further analysis research hypothesis 2 and research hypothesis 3 to explore the impact of IR applications on product upgrading and whether IR applications can weaken the motivation of China's manufacturing industry to participate in GVCs by

**Table 6. Regression results of technological heterogeneity.**

|  | (1) | (2) | (3) |
|---|---|---|---|
|  | GVCP | Forward GVCP | Backward GVCP |
| IR*technology | -0.0027*** | -0.0014*** | -0.0013** |
|  | (0.0007) | (0.0005) | (0.0007) |
| CTR | YES | YES | YES |
| Ind FE | YES | YES | YES |
| Year FE | YES | YES | YES |
| R-squared | 0.9923 | 0.9819 | 0.9553 |
| Observations | 117 | 117 | 117 |

Note

***$p < 0.01$

**$p < 0.05$, and

*$p < 0.1$; Robust standard error in parentheses; IR * technology represents the Intersection term of IR applications and high-tech/low-tech manufacturing virtual variables.

promoting product upgrading. Specifically, Eq (2) tests the impact of IR applications on product upgrading; testing Eq (1), Eq (2) and Eq (3) in turn can explain whether product upgrading plays a mediation effect between IR applications and GVCP. Table 7 reports the test results of research hypothesis 2 and research hypothesis 3.

As shown in Column (1) of Table 7, the estimated coefficient of IR application is positive, passing the 5% significance test. This indicates that the higher the level of IR applications in the industry is, the more product upgrading is promoted, which validates the research hypothesis 2 in this paper.

Column (2) reports the impact of IR applications and product upgrading on GVCP, and as can be seen, the estimated coefficients of product upgrading in the regression are all significantly negative. Combined with the calculated results in Column (1), one can learn that IR applications promote product upgrading, thereby weakening the motivation of China's

**Table 7. Regression results of research hypothesis 2 and research hypothesis 3.**

|  | (1) | (2) | (3) | (4) |
|---|---|---|---|---|
|  | Upgrade | GVCP | Forward GVCP | Backward GVCP |
| IR | 0.0112** | -0.0023*** | -0.0011*** | -0.0012* |
|  | (0.0049) | (0.0007) | (0.0004) | (0.0007) |
| Upgrade |  | -0.0351*** | -0.0250* | -0.0101** |
|  |  | (0.0118) | (0.0144) | (0.0043) |
| CLR | YES | YES | YES | YES |
| Ind FE | YES | YES | YES | YES |
| Year FE | YES | YES | YES | YES |
| R-squared | 0.9909 | 0.9931 | 0.9825 | 0.954 |
| Observations | 117 | 117 | 117 | 117 |
| Mediation Effect |  | 14.6% | 20.0% | 8.70% |

Note

***$p < 0.01$

**$p < 0.05$, and

*$p < 0.1$; Robust standard error in parentheses.

manufacturing industry to participation in GVCs. Drawing on MacKinnon et al. [53], since both β1 and γ2 have passed the significance test, Sobel test is not required. Calculate according to Eq (2), the mediation effect accounting for 14.6% of the total effect, thus verifying that IR applications have reduced in the overall participation in GVCs of China's manufacturing industry through product upgrading.

Columns (3) and (4) report the effects of IR applications and product upgrading on forward and backward participation in GVCs of China's manufacturing industry, respectively. The estimated coefficients of product upgrading are significantly negative in both Columns (3) and (4), which, combined with the calculated results in Column (1), also indicate that IR applications reduce the forward and backward participation in GVCs of China's manufacturing industry by promoting product upgrading. Specifically, the mediation effect of product upgrading on forward and backward participation in GVCs of China's manufacturing industry is 20.0% and 8.70%, respectively.

At this point, the research hypothesis 3 proposed in this paper is verified, i.e., the argument can be made that IR applications can enable product upgrading in China, which will effectively bring about a decrease in the forward participation and backward participation in GVCs of China's manufacturing industry. This, in turn, leads to a reduction in the overall participation in GVCs of China's manufacturing industry.

## Conclusion

This paper uses data from 13 manufacturing industries in China from 2006 to 2014 to empirically test the impact of IR applications on forward and backward participation in the GVCs of China's manufacturing industry. The regression results of the two-way fixed effects model indicate that IR applications lead to a simultaneous decrease in forward and backward participation in the GVCs of China's manufacturing industry. Using the 2SLS model to perform endogeneity tests on the basic regression results, it was found that the regression results of the two-way fixed effects model has good robustness. Furthermore, using the moderating effect model, the technology heterogeneity of manufacturing industry was included in the regression analysis. The empirical results showed that the application of IR suppressed both forward and backward participation in the GVCs of China's manufacturing industry, whether in high-tech or low-tech industries.

Based on the mediation effect model, this paper explains the reasons for the decline of China's manufacturing GVCP caused by IR applications from the perspective of product upgrading. IR applications promote product upgrading of China's manufacturing industry, making China more inclined to retrieve or retain manufacturing. This may have two impacts: (i) Product upgrading can help China achieve the import substitution of intermediate inputs and use local intermediate inputs to produce exports. This will reduce backward participation in the GVCs of China's manufacturing industry. (ii) Localization of manufacturing can make an economy. However, the localization of manufacturing can cause China to lose the opportunity to export intermediate inputs to other economies, thereby reducing forward participation in the GVCs.

Based on empirical research results, this paper proposes the following policy recommendations.

(i) Although the current application of IR in China has not promoted forward participation in the GVCs, the experience of developed countries shows that IR applications can help achieve an increase in forward participation. This means that China needs to strengthen the research and development of AI technology and tackle key problems, and fully leverage the empowering effect of "IR+" with a scenario driven approach to produce new products with

international competitive advantages, promote China's deep integration into the GVCs through participatory approaches.

(ii) The application of IR did not promote backward participation in the GVCs of China's manufacturing industry, which may be related to the obvious hierarchical characteristics of regional economic development within China. China needs to continue to promote the digital and intelligent transformation of its manufacturing industry, continuously enhance the added value of production and manufacturing links, promote the gradient transfer of production and manufacturing between different regions in the country, and more actively undertake the production and manufacturing activities of multinational enterprises.

(iii) The inhibitory effect of IR applications on forward and backward participation in the GVCs of China's manufacturing industry may be related to the shortage of highly skilled workers in the Chinese labor market. 'Machine substitution' does not mean that workers are no longer important, but rather that the demand for labor skills is increasing. Germany's development experience shows that highly skilled workers are an important advantage for 'Made in Germany' to participate in international competition. The Chinese government should actively learn from Germany's vocational education system and accelerate the adjustment of labor force to adapt to the changes brought about by the application of IR, which may promote the participation of China's manufacturing industry in the GVCs to move upwards.

Other avenues of research are possible. Subsequent research will adopt case study method, selecting several typical Chinese manufacturing enterprises as cases, and exploring the deep-seated reasons why the application of IR has led to a decrease in both forward and backward participation in the GVCs of Chinese manufacturing from two dimensions of theoretical construction and theoretical testing, in order to further enrich empirical research conclusions.

## Author Contributions

**Conceptualization:** Shuangzhi Zhang.

**Data curation:** Shuangzhi Zhang.

**Formal analysis:** Shuangzhi Zhang.

**Funding acquisition:** Shuangzhi Zhang.

**Investigation:** Shuangzhi Zhang.

**Methodology:** Shuangzhi Zhang.

**Project administration:** Shuangzhi Zhang.

**Resources:** Shuangzhi Zhang.

**Software:** Shuangzhi Zhang.

**Supervision:** Shuangzhi Zhang.

**Validation:** Shuangzhi Zhang.

**Visualization:** Shuangzhi Zhang.

**Writing – original draft:** Shuangzhi Zhang.

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
