## [Decision Letter · Decision Letter 0]

4 Jun 2023

PONE-D-23-12414The Impact of Industrial Robot Applications on Global Value Chain Participation of China Manufacturing Industry: Mediation Effect Based on Product UpgradingPLOS ONE

Dear Dr. zhang,

Thank you for submitting your manuscript to PLOS ONE. After careful consideration, we feel that it has merit but does not fully meet PLOS ONE’s publication criteria as it currently stands. Therefore, we invite you to submit a revised version of the manuscript that addresses the points raised during the review process.

We look forward to receiving your revised manuscript.

Kind regards,

Essossinam Ali, Ph.D

Academic Editor

PLOS ONE

Additional Editor Comments:

Although reviewers 1 and 3 give positive outcomes, I agree on many points with reviewer 2.

For example, GVC should be an in percentage. The authors are advised to divide the actual GVC by the gross export (GVC participation is equal to the sum of forward and backward participation, divided by the gross export). then run the regressions again.

The differences between industries with different technological levels should be controlled.

Use the following recent articles to strengthen the background of your paper

Ali, E., Gniniguè, M. (2022). Global value chains participation and structural transformation in Africa: Are we advocating environmental protection? Journal of Cleaner Production, 366: 132924. https://doi.org/10.1016/j.jclepro.2022.132914

Ali, E., Gniniguè, M. Awade, E.N. (2023). Sectoral value chains and environmental pollution in Africa: Can development policies target digitalization and structural transformation to enhance environmental governance? Journal of Environmental Economics and Policy, https://doi.org/10.1080/21606544.2022.2110163

Ali, E., Bataka, H.; Wonyra, K.O., Awade, E.N., Braly, N.N. (2022). Global value chains participation and environmental pollution in developing countries: Does digitalization matter?

Reviewers' comments:

Reviewer's Responses to Questions

**Comments to the Author**

1. Is the manuscript technically sound, and do the data support the conclusions?

Reviewer #1: Partly

Reviewer #2: No

Reviewer #3: Yes

2. Has the statistical analysis been performed appropriately and rigorously? 

Reviewer #1: No

Reviewer #2: No

Reviewer #3: Yes

3. Have the authors made all data underlying the findings in their manuscript fully available?

Reviewer #1: No

Reviewer #2: Yes

Reviewer #3: No

4. Is the manuscript presented in an intelligible fashion and written in standard English?

Reviewer #1: No

Reviewer #2: Yes

Reviewer #3: Yes

5. Review Comments to the Author

Reviewer #1: Dear Author,

I very much appreciate your manuscript submitted to the Journal “PLOS ONE” on The Impact of Industrial Robot Applications on Global Value Chain Participation of China Manufacturing Industry: Mediation Effect Based on Product Upgrading.

This issue is interesting and your manuscript shows your in-depth knowledge of the literature. Here are some comments that I hope will help you improve the quality of your work.

1 Introduction

The contribution of the paper is unclear. You need to improve the contribution paragraph.

2 Methodology, data and estimation techniques

2.1 Model

You state that “In Equation (1), i and t represent manufacturing industry segment and year, respectively; GVCP is global value chain participation, which is the explanatory variable;…..”. What is the dependent variable is in this equation? You need to clarify this.

2.2 Data

You try to describe data, but this it’s not enough. What are the units of the data you use?

2.3 Estimation technique

First, what is the estimation technique used for the findings reported in Tables 2-3 and Table 5? This is important because a section on estimation technique is missing. Second, throughout the text, we realize that you are using the instrumental variables (IV) approach and this is a good way to deal with the endogeneity issue. However, which type of the IV approach are you using? GMM, 2SLS or 2SLS+FE/RE? This is not clearly mentioned in the first paragraph of the section 5.3.2. Third, you state “There are two main sources of endogeneity:….” This statement is not correct. You should heck further.

2.4 Diagnostic tests

Some diagnostic tests (such as correlation and multicollinearity tests and unit-root test) are needed to support your findings. This is important because, if variables are not stationary at level, other techniques will be useful rather than the IV approach. Also, the inclusion of year fixed effect can capture the effect of some observable variables. Kindly refer some papers like https://doi.org/10.1007/s10644-023-09503-x ; https://doi.org/10.1111/ecot.12303.

3 Empirical results and analysis/Findings and discussion

3.1 Tables of results

It is difficult to understand your Tables. Why manufacturing robot stocks in US appear in Table 4 while this specification is missing in previous Table 2 or 3. According to me you applied traditional techniques in Table 2 (or3) and in Table 4 you applied the IV approach to deal with the endogeneity issue. If it is a case, I think that your explanatory variable of interest (IR) must be the same when you move from Table 2 (or 3) to Table 4.

In the same vein, for Table 5, you must present the findings based on traditional and IV approach methods.

3.2 Presentation of the empirical results and their discussion

The Author has separated Section 5 Empirical result and analysis from Section 6 Research findings and discussion. This manner of presentation is less recommended. According to me, Section 5 does not differ from Section 6. In the same vein, it is be useful to report the empirical result and analysis and research findings and discussion in the same section. Also, the decomposition of the Section 6 into two sub-sections 6.1 research findings and 6.2 research discussion is not necessary, this means that the findings and their discussion must be simultaneously presented.

4 Conclusion

The whole conclusion section is ignored. I recommend the author to include a conclusion section.

5 Language editing

The current version of the paper is not well edited. Grammar and spelling checks are required.

Reviewer #2: Using China manufacturing industry data, working from the perspective of product upgrading, this paper empirically analyzes the impact of IR applications on GVCP. For the following reasons, we suggest this paper to be rejected.

1. GVC participation is equal to the sum of forward and backward participation. Obviously, when IR applications could reduce forward and backward participation respectively, IR applications would inevitably reduce GVC participation. Therefore, the necessity of research from the three aspects of forward and backward participation and total participation is relatively weak.

2. Section 4.2.1 spend a lot of space to introduce the background of the value-added decomposition framework. This seems unnecessary.

3. The author does not consider the differences between industries with different technological levels.

4. Compared to product upgrading, production costs and production efficiency also play an important role in the impact of IR application on GVC participation. However, this is not analyzed in this paper.

5. It is suggested that the author revise the title of Section 4.1.1 and Section 4.1.2 to highlight the research content. The same problem also exists in Section 5.

Reviewer #3: The author examines the impact of IR applications on global value chain participation in Chinese manufacturing industry for the years 2006-2014 by focusing on the mediating effects of product upgrading.

Overall this is a nice paper, and I do not have major substantive changes, but instead a number of smaller revisions. These changes are listed below.

• The literature review is comprehensive enough and discuss the constituents of the GVC participation and industrial robots nexus deeply. However, there are too much repetition in the paper. For example, the concept of "make cage for bird" is repeated three times in the paper. To avoid the repetitive arguments and phrases throughout the manuscript, Sections 3 and 4 can be merged into more shortened section presenting hypotheses and a model.

• The conclusion that the negative impact of IR applications on GVC participation could possibly be related to the size and development levels of a country cannot be generalized and applied to other cases though because the results are presented here come from Chinese economy. This limitation should be mentioned in the text.

• GVC participation can be calculated by different methodologies such as Koopman et al (2010) (trade based) and Wang et al (2017) (production based). Here the author chooses the methodology of Koopman et al (2010). The alternative measures can be mentioned as further research directions.

• This study covers the period 2006-2014 regarding the WIOD. However, OECD TİVA 2021 continue until 2018 and OECD TiVA 2022 continues until 2020. This remaining period from 2014 to 2020 includes very significant worldwide occasions having rigorous impact on GVCs. The author may clarify that the availability of the Socio Economic Accounts (SEA) of the WIOD is superior to OECD TiVA Data especially in terms of the availability of the control variables.

• The author seems to largely ignore the role of foreign direct investment and multinationals in the discussions. The author can provide some discussion on this.

• I recommend author to control the impact of the 2008 global economic crisis by employing a dummy variable.

• Table 1 should also present the data on backward and forward GVCP individually to the impact on each of them.

• As robustness checks, in addition to the use of instrumental variable of IR in the United States, the author may also provide the estimates with IR in the EU and Japan.

6. PLOS authors have the option to publish the peer review history of their article (what does this mean?). If published, this will include your full peer review and any attached files.

Reviewer #1: No

Reviewer #2: No

Reviewer #3: No

---

## [Author Response · Author response to Decision Letter 0]

20 Jul 2023

Response to Reviewers

Thank you very much to Professor Ali and the reviewers for giving me this valuable opportunity to revise my paper. After a long time of effort, I have finally completed this revision. Now, I will provide the following explanation for the revisions. If there are any mistakes, please continue to provide criticism and assistance from all experts. We appreciate it very much.

1.Response to the academic editor

（1）GVC should be an in percentage. The authors are advised to divide the actual GVC by the gross export (GVC participation is equal to the sum of forward and backward participation, divided by the gross export). then run the regressions again.

Thank you very much for your valuable feedback. Based on your feedback, I have re examined the variable settings in this article. This paper measures the GVCP of Chinese industries using the GVCP measure proposed by Koopman et al., as shown in Equation (1).

GVCP=IV/E+FV/E (1)

In Equation (1), GVCP is the global value chain participation index; IV denotes the domestic value-added in intermediate products processed in the importing country and then exported to third-party countries; FV denotes the foreign value-added in domestic exports, and E denotes total exports. The larger the index is, the higher is the industry's participation in GVCs. Further, this participation index can be decomposed into forward and backward participation, with IV/E being forward participation and FV/E being backward participation. Forward and backward participation can indicate the degree of participation of a country's industry in the upstream and downstream links of GVCs.

In summary, the calculation method of GVCP is in line with your opinion. This article supplemented the control variables based on the review comments and re conducted regression analysis in sequence.

（2）The differences between industries with different technological levels should be controlled.

Thank you very much for your valuable feedback. This article uses binary dummy variables to distinguish between high and low tech manufacturing industries, where "1" represents high-tech manufacturing and "0" represents low-tech manufacturing. To test whether the basic regression results will be affected by different technological levels in the manufacturing industry, this article introduces moderating variables for regression analysis. By setting the intersection term between IR applications and high-tech/low-tech manufacturing virtual variables as the explanatory variable, basic equations is regressed again. The empirical results indicate that IR applications have a significant inhibitory effect on the forward and backward GVCP of China's manufacturing industry in both high-tech and low-tech industries.

（3）Use the following recent articles to strengthen the background of your paper.

Thank you very much for your valuable feedback. After carefully reading these academic papers, we have added them to the statement of research background (including literature review and research hypotheses) accordingly.

2.Response to the reviewer 1

（1）The contribution of the paper is unclear. You need to improve the contribution paragraph.

Based on your valuable feedback, we have rewritten the research contribution section.

（2）You state that “In Equation (1), i and t represent manufacturing industry segment and year, respectively; GVCP is global value chain participation, which is the explanatory variable;…..”. What is the dependent variable is in this equation? You need to clarify this.

Based on your valuable feedback, we have made corresponding modifications to this section: ‘In Equation (1), i and t represent manufacturing industry segment and year, respectively; GVCP is global value chain participation, which is the result variable; IR represents the degree of IR applications, which is the explanatory variable.’

（3）You try to describe data, but this it’s not enough. What are the units of the data you use?

Based on your valuable feedback, we have made corresponding modifications to the data description section and added annotations to Table 1. ‘Note: ①The unit of ‘robot stock’ is one robot, for example, the value of 0.01 in the food and beverages industry in 2006 represents a robot inventory of 0.01 units per 1,000 employed persons in2006. The larger the number, the higher the level of IR applications in the industry. ②The number range of GVCP is from 0 to 1, and there is no specific unit. The larger the number, the higher the GVCP representing the industry.’

（4）Estimation technique. First, what is the estimation technique used for the findings reported in Tables 2-3 and Table 5? This is important because a section on estimation technique is missing. Second, throughout the text, we realize that you are using the instrumental variables (IV) approach and this is a good way to deal with the endogeneity issue. However, which type of the IV approach are you using? GMM, 2SLS or 2SLS+FE/RE? This is not clearly mentioned in the first paragraph of the section 5.3.2. Third, you state “There are two main sources of endogeneity:….” This statement is not correct. You should heck further.

Thank you very much for your valuable feedback. We have made the following revisions in sequence.

First, the estimation technique used in Table 2, Table 3 and Table 5 (now Table 7) is Ordinary least squares (OLS).

Secondly, the estimation technique used in Table 4 is the two-stage least squares method (2SLS), which controls for time and industry fixed effects.

Finally, we have made revisions to the discussion on the reasons for endogeneity, not limited to the two reasons proposed in the original manuscript.

（5）Diagnostic tests. Some diagnostic tests (such as correlation and multicollinearity tests and unit-root test) are needed to support your findings. This is important because, if variables are not stationary at level, other techniques will be useful rather than the IV approach. Also, the inclusion of year fixed effect can capture the effect of some observable variables. Kindly refer some papers like https://doi.org/10.1007/s10644-023-09503-x ; https://doi.org/10.1111/ecot.12303.

Thank you very much for your valuable feedback. We have added corresponding diagnostic test instructions (such as correlation and multicollinearity tests and unit root tests) in the descriptive analysis section.

(i) Through the calculation of Pearson correlation coefficient, it was found that there were no values exceeding 0.40 in the three samples of GVCP, forward GVCP, and back GVCP, indicating a weak correlation between the variables.

(ii) Through the calculation of the VIF coefficient, it is found that there is no more than 2.00 values in GVCP, forward GVCP, and backward GVCP, which indicates that there is no multicollinearity problem between variables.

(iii) This paper conducts LLC test on the panel unit-root of GVCP, forward GVCP, and backward GVCP, and finds that these adjusted t_δ^* statistics are significantly negative, which means strongly rejecting the original assumption that the panel contains Root of unity and accepting the conclusion that the panel is a Stationary process. Further, after subtracting the cross-sectional mean from Panel data, the LLC test was conducted to avoid the possible cross-sectional correlation effects of the disturbance terms in different industries. The results showed that these adjusted t_δ^* statistics are still significantly negative at the 1% level.

Finally, based on the literature recommended by the reviewers, important information that may be implied by the fixed time effect was discussed, and relevant references were added to the references.

（6）Tables of results. It is difficult to understand your Tables. Why manufacturing robot stocks in US appear in Table 4 while this specification is missing in previous Table 2 or 3. According to me you applied traditional techniques in Table 2 (or3) and in Table 4 you applied the IV approach to deal with the endogeneity issue. If it is a case, I think that your explanatory variable of interest (IR) must be the same when you move from Table 2 (or 3) to Table 4. In the same vein, for Table 5, you must present the findings based on traditional and IV approach methods.

Thank you very much for your valuable comments. The number of robots stocked per 1000 employed population in the related industries in the United States is taken as a Instrumental variables estimation in the columns (1) to (3) of Table 4, and the regression analysis of equation (1) is conducted again using the 2SLS method. Columns (4) to (6) of Table 4 use the lag phase I of the explanatory variable as a Instrumental variables estimation, and use OLS method to conduct regression analysis on equation (1) again. Table 5 (now Table 7) follows the step by step regression steps of Mesomeric effect, and tests the Mesomeric effect of product upgrading. OLS technology is used for regression analysis. However, whether the stepwise regression analysis using the 2SLS method can test the Mesomeric effect, the relevant literature we have (for example, MacKinnon et al. [51], Selig and Preacher [52], and Zhao et al. [53]) does not carry out a robustness test on the Mesomeric effect. So, we are currently unable to solve this issue effectively. Please understand and thank you very much.

（7）Presentation of the empirical results and their discussion. The Author has separated Section 5 Empirical result and analysis from Section 6 Research findings and discussion. This manner of presentation is less recommended. According to me, Section 5 does not differ from Section 6. In the same vein, it is be useful to report the empirical result and analysis and research findings and discussion in the same section. Also, the decomposition of the Section 6 into two sub-sections 6.1 research findings and 6.2 research discussion is not necessary, this means that the findings and their discussion must be simultaneously presented.

Thank you very much for your valuable feedback. We have made significant deletions to the last part of the original manuscript and have rewritten it.

（8）Conclusion. The whole conclusion section is ignored. I recommend the author to include a conclusion section.

Thank you very much for your valuable feedback. We have rewritten the conclusion section of this article based on the modification suggestions mentioned in (7) above.

（9）Language editing. The current version of the paper is not well edited. Grammar and spelling checks are required.

Thank you very much for your valuable feedback. We have selected PLOS ONE's language polishing service to fully revise the manuscript language.

3.Response to the reviewer 2

（1）Section 4.2.1 spend a lot of space to introduce the background of the value-added decomposition framework. This seems unnecessary.

Thank you very much for your valuable feedback. There are many methods for measuring GVCP, and we tend to measure this variable from the perspective of value-added decomposition. Therefore, we have devoted more space to explaining the calculation of GVCP.

（2）The author does not consider the differences between industries with different technological levels.

Thank you very much for your valuable feedback. We have added the 'Analysis of Technical Heterogeneity' section to analyze the technical heterogeneity of the basic regression results.

（3）Compared to product upgrading, production costs and production efficiency also play an important role in the impact of IR application on GVC participation. However, this is not analyzed in this paper.

Thank you very much for your valuable feedback. As this article mainly aims to illustrate that product upgrading is an important factor affecting the decline of the global value chain in China's manufacturing industry, it mainly focuses on product upgrading for discussion. Based on your revision comments, we have provided a theoretical explanation of the other variables that may have an impact that you proposed in the literature review section.

（4）It is suggested that the author revise the title of Section 4.1.1 and Section 4.1.2 to highlight the research content. The same problem also exists in Section 5.

Thank you very much for your valuable feedback. We have made modifications to the corresponding titles.

4.Response to the reviewer 3

（1）The literature review is comprehensive enough and discuss the constituents of the GVC participation and industrial robots nexus deeply. However, there are too much repetition in the paper. For example, the concept of "make cage for bird" is repeated three times in the paper. To avoid the repetitive arguments and phrases throughout the manuscript, Sections 3 and 4 can be merged into more shortened section presenting hypotheses and a model.

Thank you very much for your valuable feedback. We have edited and improved the literature review and theoretical assumptions section of the manuscript.

（2）The conclusion that the negative impact of IR applications on GVC participation could possibly be related to the size and development levels of a country cannot be generalized and applied to other cases though because the results are presented here come from Chinese economy. This limitation should be mentioned in the text.

Thank you very much for your valuable feedback. We have explained the limitations of the research in the abstract, introduction, research conclusion, and other parts of the manuscript. The research conclusion of the manuscript is only applicable to the Chinese sample.

（3）GVC participation can be calculated by different methodologies such as Koopman et al (2010) (trade based) and Wang et al (2017) (production based). Here the author chooses the methodology of Koopman et al (2010). The alternative measures can be mentioned as further research directions.

Thank you very much for your valuable feedback. As the data we currently have mainly involves trade value-added, if we have suitable data in the future, we will try to calculate GVCP from the perspective of production, in order to obtain some valuable research conclusions.

（4）This study covers the period 2006-2014 regarding the WIOD. However, OECD TİVA 2021 continue until 2018 and OECD TiVA 2022 continues until 2020. This remaining period from 2014 to 2020 includes very significant worldwide occasions having rigorous impact on GVCs. The author may clarify that the availability of the Socio Economic Accounts (SEA) of the WIOD is superior to OECD TiVA Data especially in terms of the availability of the control variables.

Thank you very much for your valuable feedback. As the 2016 version of WIOD is currently the latest publicly available data, we regret to set the empirical period to 2006-2014. Although we are currently unable to address the issue of data timeliness, the research conclusion is quite relevant to the Chinese context, that is, the global value chain of China's manufacturing industry is indeed showing a downward trend, which is consistent with the transfer of labor-intensive industries from the eastern coastal areas of China to the central and western regions, as well as the current construction of domestic circular markets. In the future, on the one hand, we actively collect the latest data for regression analysis, and on the other hand, through Case study of Chinese enterprises, we hope to further expand this study. I hope our answer can receive your understanding. Thank you very much.

（5）The author seems to largely ignore the role of foreign direct investment and multinationals in the discussions. The author can provide some discussion on this.

Thank you very much for your valuable feedback. Due to data limitations, we cannot include variables such as foreign direct investment in the control variables. But based on your opinion, we have added relevant literature to the literature review and theoretical hypothesis section for theoretical explanation.

（6）I recommend author to control the impact of the 2008 global economic crisis by employing a dummy variable.

Based on your revision comments, we have added dummy variables from the 2008 financial crisis to the control variables and conducted a new regression analysis.

（7）Table 1 should also present the data on backward and forward GVCP individually to the impact on each of them.

Based on your revision comments, we have added data descriptions of forward and backward participation in the global value chain of China's manufacturing industry in Table 1. Thank you very much for your valuable feedback.

（8）As robustness checks, in addition to the use of instrumental variable of IR in the United States, the author may also provide the estimates with IR in the EU and Japan.

Thank you very much for your valuable comments. In the Instrumental variables estimation section of the manuscript, the reason for using US data is explained, that is, US data meets exogenous and relevance requirements, and is an ideal Instrumental variables estimation. Considering that China, along with India, Brazil, South Africa, and Russia, are emerging economies with faster development rates, corresponding regression analysis can be conducted using data from these countries, and the basic conclusions have been verified. Due to some missing data from South Africa, we only returned data from India, Brazil, and Russia, as shown in the table below. The research results indicate that the application of industrial robots significantly suppressed the overall and backward participation in the global value chain of manufacturing in these developing countries. Although the forward participation did not pass the significance test, the regression coefficient was negative.

Table1 Based on regression results for India, Brazil, and Russia

 (1)

GVCP (2)

Forward GVCP (3)

Backward GVCP

IR -0.0089*** -0.0009 -0.0080***

 (0.0019) (0.0018) (0.0022)

CTR YES YES YES

Ind FE YES YES YES

Year FE YES YES YES

Country FE YES YES YES

R-squared 0.7592 0.8394 0.6210

Observations 570 570 570

Note: ***p < 0.01, **p < 0.05, and *p < 0.1; Robust standard error in parentheses.

---

## [Decision Letter · Decision Letter 1]

25 Aug 2023

PONE-D-23-12414R1The Impact of Industrial Robot Applications on Global Value Chain Participation of China Manufacturing Industry: Mediation Effect Based on Product UpgradingPLOS ONE

Dear Dr. zhang,

Thank you for submitting your manuscript to PLOS ONE. After careful consideration, we feel that it has merit but does not fully meet PLOS ONE’s publication criteria as it currently stands. Therefore, we invite you to submit a revised version of the manuscript that addresses the points raised during the review process.

ACADEMIC EDITOR:

Dear Authors,

Thank you for revising your manuscript. While the paper has a potential contribution to the literature, you need much more work on it.

1. Please take seriously the concerns raised by reviewer 1. I Checked the paper and, fully agreed with him.

2. The following paper can strengthen the bacground of your research

Ali, E., Bataka, H.; Wonyra, K.O., Awade, E.N., **Braly, N.N**. (2023). Global value chains participation and environmental pollution in developing countries : Does digitalization matter? **Journal of International Development**, http://doi.org/10.1002/jid.3823

.==============================

We look forward to receiving your revised manuscript.

Kind regards,

Essossinam Ali, Ph.D

Academic Editor

PLOS ONE

Additional Editor Comments:

Dear Authors,

Thank you for revising your manuscript. While the paper has a potential contribution to the literature, you need much more work on it.

1. Please take seriously the concerns raised by reviewer 1. I Checked the paper and, fully agreed with him.

2. The following paper can strengthen your research background.

Ali, E., Bataka, H.; Wonyra, K.O., Awade, E.N., Braly, N.N. (2023). Global value chains participation and environmental pollution in developing countries : Does digitalization matter? Journal of International Development, http://doi.org/10.1002/jid.3823

Reviewers' comments:

Reviewer's Responses to Questions

**Comments to the Author**

1. If the authors have adequately addressed your comments raised in a previous round of review and you feel that this manuscript is now acceptable for publication, you may indicate that here to bypass the “Comments to the Author” section, enter your conflict of interest statement in the “Confidential to Editor” section, and submit your "Accept" recommendation.

Reviewer #1: (No Response)

Reviewer #3: All comments have been addressed

2. Is the manuscript technically sound, and do the data support the conclusions?

Reviewer #1: Partly

Reviewer #3: Yes

3. Has the statistical analysis been performed appropriately and rigorously? 

Reviewer #1: Yes

Reviewer #3: Yes

4. Have the authors made all data underlying the findings in their manuscript fully available?

Reviewer #1: No

Reviewer #3: No

5. Is the manuscript presented in an intelligible fashion and written in standard English?

Reviewer #1: Yes

Reviewer #3: Yes

6. Review Comments to the Author

Reviewer #1: Dear Author,

I very much appreciate your revised manuscript submitted to the Journal “PLOS ONE” on The Impact of Industrial Robot Applications on Global Value Chain Participation of China Manufacturing Industry: Mediation Effect Based on Product Upgrading.

This issue is interesting and your manuscript shows your in-depth knowledge of the literature. Here are some comments that I hope will help you improve the quality of your work.

1- The author has discussed some diagnostic tests as previously requested. However, this discussion needs to be supported by some previous studies. For some of these tests, again, kindly refer to some recent papers like https://doi.org/10.1007/s10644-023-09503-x ; https://doi.org/10.1016/j.strueco.2022.10.003 ; https://doi.org/10.1111/ecot.12303.

2- Through the text, some abbreviations, like LLC, need to be defined first.

3- The conclusion section must be revised.

(i) Some important points like the methodology applied and the main findings should be provided in the conclusion.

(ii) Policy implications should flow directly from your findings.

(iii) Each study has its strengths and weaknesses (limitations). Thus, the author should provide avenues for future research.

Reviewer #3: 2nd round review on “The Impact of Industrial Robot Applications on Global Value Chain Participation of China Manufacturing Industry: Mediation Effect Based on Product Upgrading.”

Based on our earlier feedback, the manuscript has been improved with the substantial effort by the author. The author has revised the literature review in more comprehensive style and removed the repetitive arguments. The limitations of the study are well articulated based on our earlier review in the manuscript. Also, the robustness check seems to validate the earlier empirical results. But some minor points in the manuscript should be addressed.

-It is suggested that the review includes the theoretical underpinnings linking the adoption of industrial robotics to the emergence of new skills and job profiles. The adoption of robotics may lead to the creation of new job profiles/skills, like robot maintenance or programming, while simultaneously reducing demand for low-skilled manual jobs. Countries that can quickly adapt their workforce to these changes can potentially move up the GVC ladder. Including these theoretical insights would enhance the depth of the review.

7. PLOS authors have the option to publish the peer review history of their article (what does this mean?). If published, this will include your full peer review and any attached files.

Reviewer #1: No

Reviewer #3: No

---

## [Author Response · Author response to Decision Letter 1]

30 Aug 2023

Response to Reviewers

Thank you very much to Professor Ali and the reviewers for giving me this valuable opportunity to revise this paper. Now, we will provide the following explanation for the revisions. If there are any mistakes, we sincerely hope to continue to receive guidance from Professor Ali and the reviewers.

Kind regards,

Shuangzhi Zhang

Data Availability Statement: All relevant data are within the manuscript and its Supporting information files.

The dataset of this paper has been uploaded as an attachment in the submission system and has been uploaded to the Dryad database.

DOI: 10.5061/dryad.280gb5mvr

URL: https://datadryad.org/stash/share/ojZHg4V2fxznQ402yT_jo6EkP4fFblF2co_d5iI0VJM.

1.Response to the Academic Editor

（1）Please take seriously the concerns raised by reviewer 1. I Checked the paper and, fully agreed with him.

Thank you very much for your valuable feedback. After carefully reading the comments of Reviewer 1, we made careful revisions one by one. The specific situation can be seen in the response to Reviewer 1's comments.

（2）The following paper can strengthen your research background.

Thank you very much for providing these important references. After carefully reading this paper, we will use it to expand the content of the literature review. We have added it as reference [33].

2.Response to the Reviewer 1

（1）The author has discussed some diagnostic tests as previously requested. However, this discussion needs to be supported by some previous studies. For some of these tests, again, kindly refer to some recent papers like https://doi.org/10.1007/s10644-023-09503-x ; https://doi.org/10.1016/j.strueco.2022.10.003 ; https://doi.org/10.1111/ecot.12303.

Thank you very much for providing these important references. After carefully reading these papers, we will use them as an important basis for the research background and empirical model selection of this paper. We have added them as references [61], [65] and [67].

（2）Through the text, some abbreviations, like LLC, need to be defined first.

Thank you very much for your constructive feedback. Based on your valuable feedback, we have provided an explanation and full name description of the first appearance of LLC. We have also carefully reviewed the entire text to avoid similar errors.

（3）The conclusion section must be revised.

(i) Some important points like the methodology applied and the main findings should be provided in the conclusion.

(ii) Policy implications should flow directly from your findings.

(iii) Each study has its strengths and weaknesses (limitations). Thus, the author should provide avenues for future research.

Based on your valuable feedback, we have rewritten the conclusion section, which is very important for improving the quality of this paper. The specific content is as follows:

This paper uses data from 13 manufacturing industries in China from 2006 to 2014 to empirically test the impact of IR applications on forward and backward participation in the GVCs of China's manufacturing industry. The regression results of the two-way fixed effects model indicate that IR applications lead to a simultaneous decrease in forward and backward participation in the GVCs of China's manufacturing industry. Using the 2SLS model to perform endogeneity tests on the basic regression results, it was found that the regression results of the two-way fixed effects model has good robustness. Furthermore, using the moderating effect model, the technology heterogeneity of manufacturing industry was included in the regression analysis. The empirical results showed that the application of IR suppressed both forward and backward participation in the GVCs of China's manufacturing industry, whether in high-tech or low-tech industries.

Based on the mediation effect model, this paper explains the reasons for the decline of China's manufacturing GVCP caused by IR applications from the perspective of product upgrading. IR applications promote product upgrading of China's manufacturing industry, making China more inclined to retrieve or retain manufacturing. This may have two impacts: (i) Product upgrading can help China achieve the import substitution of intermediate inputs and use local intermediate inputs to produce exports. This will reduce backward participation in the GVCs of China's manufacturing industry. (ii) Localization of manufacturing can make an economy. However, the localization of manufacturing can cause China to lose the opportunity to export intermediate inputs to other economies, thereby reducing forward participation in the GVCs.

Based on empirical research results, this paper proposes the following policy recommendations.

(i) Although the current application of IR in China has not promoted forward participation in the GVCs, the experience of developed countries shows that IR applications can help achieve an increase in forward participation. This means that China needs to strengthen the research and development of AI technology and tackle key problems, and fully leverage the empowering effect of "IR+" with a scenario driven approach to produce new products with international competitive advantages, promote China's deep integration into the GVCs through participatory approaches.

(ii) The application of IR did not promote backward participation in the GVCs of China's manufacturing industry, which may be related to the obvious hierarchical characteristics of regional economic development within China. China needs to continue to promote the digital and intelligent transformation of its manufacturing industry, continuously enhance the added value of production and manufacturing links, promote the gradient transfer of production and manufacturing between different regions in the country, and more actively undertake the production and manufacturing activities of multinational enterprises.

(iii) The inhibitory effect of IR applications on forward and backward participation in the GVCs of China's manufacturing industry may be related to the shortage of highly skilled workers in the Chinese labor market. 'Machine substitution' does not mean that workers are no longer important, but rather that the demand for labor skills is increasing. Germany's development experience shows that highly skilled workers are an important advantage for 'Made in Germany' to participate in international competition. The Chinese government should actively learn from Germany's vocational education system and accelerate the adjustment of labor force to adapt to the changes brought about by the application of IR, which may promote the participation of China's manufacturing industry in the GVCs to move upwards.

Other avenues of research are possible. Subsequent research will adopt case study method, selecting several typical Chinese manufacturing enterprises as cases, and exploring the deep-seated reasons why the application of IR has led to a decrease in both forward and backward participation in the GVCs of Chinese manufacturing from two dimensions of theoretical construction and theoretical testing, in order to further enrich empirical research conclusions.

3.Response to the Reviewer 3

It is suggested that the review includes the theoretical underpinnings linking the adoption of industrial robotics to the emergence of new skills and job profiles. The adoption of robotics may lead to the creation of new job profiles/skills, like robot maintenance or programming, while simultaneously reducing demand for low-skilled manual jobs. Countries that can quickly adapt their workforce to these changes can potentially move up the GVC ladder. Including these theoretical insights would enhance the depth of the review.

Thank you very much for your valuable feedback. Your opinion provides a very important perspective for improving the argumentation height of this paper. Based on your opinion, we have attempted to supplement the conclusion section, and the specific content is as follows.

The inhibitory effect of IR applications on forward and backward participation in the GVCs of China's manufacturing industry may be related to the shortage of highly skilled workers in the Chinese labor market. 'Machine substitution' does not mean that workers are no longer important, but rather that the demand for labor skills is increasing. Germany's development experience shows that highly skilled workers are an important advantage for 'Made in Germany' to participate in international competition. The Chinese government should actively learn from Germany's vocational education system and accelerate the adjustment of labor force to adapt to the changes brought about by the application of IR, which may promote the participation of China's manufacturing industry in the GVCs to move upwards.

---

## [Decision Letter · Decision Letter 2]

2 Oct 2023

PONE-D-23-12414R2The Impact of Industrial Robot Applications on Global Value Chain Participation of China Manufacturing Industry: Mediation Effect Based on Product UpgradingPLOS ONE

Dear Dr. zhang, 

Thank you for submitting your manuscript to PLOS ONE. After careful consideration, we feel that it has merit but does not fully meet PLOS ONE’s publication criteria as it currently stands. Therefore, we invite you to submit a revised version of the manuscript that addresses the points raised during the review process.

Kindly review the journal guidline and check the references styles. Please use the APA style for all references cited in the reference list

Please submit your revised manuscript by Nov 16 2023 11:59PM. If you will need more time than this to complete your revisions, please reply to this message or contact the journal office at plosone@plos.org. Please include the following items when submitting your revised manuscript:A rebuttal letter that responds to each point raised by the academic editor and reviewer(s). You should upload this letter as a separate file labeled 'Response to Reviewers'.A marked-up copy of your manuscript that highlights changes made to the original version. You should upload this as a separate file labeled 'Revised Manuscript with Track Changes'.An unmarked version of your revised paper without tracked changes. You should upload this as a separate file labeled 'Manuscript'.If applicable, we recommend that you deposit your laboratory protocols in protocols.io to enhance the reproducibility of your results. Protocols.io assigns your protocol its own identifier (DOI) so that it can be cited independently in the future. For instructions see: https://journals.plos.org/plosone/s/submission-guidelines#loc-laboratory-protocols. Additionally, PLOS ONE offers an option for publishing peer-reviewed Lab Protocol articles, which describe protocols hosted on protocols.io. Read more information on sharing protocols at https://plos.org/protocols?utm_medium=editorial-email&utm_source=authorletters&utm_campaign=protocols.

We look forward to receiving your revised manuscript.

Kind regards,

Essossinam Ali, Ph.D

Academic Editor

PLOS ONE

Journal Requirements:

**Additional Editor Comments:**

Dear Authors,

Review the journal guidline and check the references styles. Please use the APA style for all references cited in the reference list

Thank you

Reviewers' comments:

Reviewer's Responses to Questions

**Comments to the Author**

1. If the authors have adequately addressed your comments raised in a previous round of review and you feel that this manuscript is now acceptable for publication, you may indicate that here to bypass the “Comments to the Author” section, enter your conflict of interest statement in the “Confidential to Editor” section, and submit your "Accept" recommendation.

Reviewer #1: All comments have been addressed

2. Is the manuscript technically sound, and do the data support the conclusions?

Reviewer #1: Yes

3. Has the statistical analysis been performed appropriately and rigorously? 

Reviewer #1: Yes

4. Have the authors made all data underlying the findings in their manuscript fully available?

Reviewer #1: No

5. Is the manuscript presented in an intelligible fashion and written in standard English?

Reviewer #1: Yes

6. Review Comments to the Author

Reviewer #1: Congratulations, the author(s) has(ve) fully taken into account our previous comments.

As additional comment, you should harmonize your references and respect the order these references. For example, the author(s) should review certain references that are incorrectly entered, particularly references beginning with Essotanam or Essossinam (examples are [8], [32], [33], [61], [65], [67]). Therefore, in the section of references, the first names of the authors must be abbreviated.

7. PLOS authors have the option to publish the peer review history of their article (what does this mean?). If published, this will include your full peer review and any attached files.

Reviewer #1: No

---

## [Author Response · Author response to Decision Letter 2]

9 Oct 2023

Response to Reviewers

Dear Professor Ali and Reviewers:

Thank you very much to Professor Ali and the reviewers for recognizing the revision work of this paper, which is very important to me. It is precisely based on your constructive suggestions that this paper can have a significant improvement in quality. Based on the suggestions of the reviewers and the journal guidelines, I have made very detailed revisions to the references in this paper and marked them in the text.

I would like to express my great appreciation to you and the reviewers for their comments on our paper. I look forward to hearing from you.

Thank you, and best regards.

Yours sincerely,

ShuangZhi Zhang 

Data Availability Statement: All relevant data are within the manuscript and its Supporting information files. The dataset of this paper has been uploaded as an attachment in the submission system and has been uploaded to the Dryad database.

DOI: 10.5061/dryad.280gb5mvr

URL: https://datadryad.org/stash/share/ojZHg4V2fxznQ402yT_jo6EkP4fFblF2co_d5iI0VJM

Figure Files: According to the journal's paper submission guidelines, all charts in this article have been uploaded to the Preflight Analysis and Conversion Engine (PACE) platform.

1.Response to the Academic Editor

Kindly review the journal guidline and check the references styles. Please use the APA style for all references cited in the reference list.

Thank you very much for your valuable feedback. According to the journal's paper submission guidelines, I have made very detailed revisions to the references for this paper and marked them in the text. Please refer to the revised references, I hope that you will be satisfied with my revision.

2.Response to the Reviewer #1

Congratulations, the author(s) has(ve) fully taken into account our previous comments.

As additional comment, you should harmonize your references and respect the order these references. For example, the author(s) should review certain references that are incorrectly entered, particularly references beginning with Essotanam or Essossinam (examples are [8], [32], [33], [61], [65], [67]). Therefore, in the section of references, the first names of the authors must be abbreviated.

Thank you very much for your recognition of the revision work of this paper, which is very important to me. Based on your constructive feedback and journal guidelines, I have carefully revised the references for this paper. Please refer to the revised references, I hope that you will be satisfied with my revision.

---

## [Editor Report · Decision Letter 3]

12 Oct 2023

The Impact of Industrial Robot Applications on Global Value Chain Participation of China Manufacturing Industry: Mediation Effect Based on Product Upgrading

PONE-D-23-12414R3Dear Authors,We are pleased to inform you that your manuscript has been judged scientifically suitable for publication and will be formally accepted for publication once it meets all outstanding technical requirements.Within one week, you will receive an e-mail detailing the required amendments. When these have been addressed, you will receive a formal acceptance letter and your manuscript will be scheduled for publication.

Kind regards,

Essossinam Ali, Ph.D

Academic Editor

PLOS ONE

Additional Editor Comments (optional):

Congratulations in revising your manuscript. Please, during the proofreading, kindly delete "The" at the beginning of the title of the article.
---

## [Editor Report · Acceptance letter]

19 Oct 2023

PONE-D-23-12414R3 

Impact of Industrial Robot Applications on Global Value Chain Participation of China Manufacturing Industry: Mediation Effect Based on Product Upgrading 

Dear Dr. Zhang:

I'm pleased to inform you that your manuscript has been deemed suitable for publication in PLOS ONE. Congratulations! Your manuscript is now with our production department. 

Kind regards, 

on behalf of

Dr. Essossinam Ali 

Academic Editor

PLOS ONE